# Risk Factors and Ocular Health Associated with Toxoplasmosis in *Quilombola* Communities

**DOI:** 10.3390/microorganisms14010096

**Published:** 2026-01-01

**Authors:** Silvio Carneiro Cunha Filho, Sandro Estevan Moron, Raphael Gomes Ferreira, Helierson Gomes, Noé Mitterhofer Eiterer Ponce de Leon da Costa, Alex Sander Rodrigues Cangussu, Bergmann Morais Ribeiro, Fabricio Souza Campos, Gil Rodrigues dos Santos, Raimundo Wagner de Souza Aguiar, Thaís Ribeiro Costa, Elainy Cristina Alves Martins Oliveira, Julliana Dias Pinheiro, Frederico Eugênio, Erica Eugênio Lourenço Gontijo, Sara Falcão de Sousa, Marcos Gontijo da Silva

**Affiliations:** 1Medical School, Health Sciences Center, Federal University of Northern Tocantins (UFNT), Araguaína 77814-350, Tocantins, Brazil; silvio_carneiro@mail.uft.edu.br (S.C.C.F.); sandromoron@mail.uft.edu.br (S.E.M.); ovraphael@gmail.com (R.G.F.); helierson.gomes@ufnt.edu.br (H.G.); 2Graduate Program in Biotechnology, Federal University of Tocantins, Gurupi 77402-970, Tocantins, Brazil; noe.eiterer@mail.uft.edu.br (N.M.E.P.d.L.d.C.); alexcangussu@uft.edu.br (A.S.R.C.); bergmannribeiro@gmail.com (B.M.R.); camposvet@gmail.com (F.S.C.); gilrsan@uft.edu.br (G.R.d.S.); rwsa@mail.uft.edu.br (R.W.d.S.A.); ribeiro.thais@mail.uft.edu.br (T.R.C.); biocris@uft.edu.br (E.C.A.M.O.); 3Gurupi Regional Hospital, Gurupi 77405-110, Tocantins, Brazil; jullianapinheiro@unirg.edu.br (J.D.P.); fredgpi@hotmail.com (F.E.); 4Graduate Program in Biosciences and Health at UNIRG, Gurupi 77425-500, Tocantins, Brazil; ericagontijo@unirg.edu.br (E.E.L.G.); sarafalcao@unirg.edu.br (S.F.d.S.)

**Keywords:** toxoplasmosis, risk factors, blindness, *Quilombolas*, *Toxoplasma gondii*, *Quilombola* communities

## Abstract

Toxoplasmosis is a parasitic disease associated with significant morbidity and mortality. This cross-sectional study aimed to determine the prevalence, associated risk factors, and ocular health outcomes related to *Toxoplasma gondii* seropositivity in 161 residents from four *Quilombolas* communities in the northern region of Tocantins, Brazilian Legal Amazon. Peripheral blood samples were collected and tested by Enzyme-Linked Immunosorbent Assay (ELISA) for Immunoglobulin G (IgG) and/or Immunoglobulin M (IgM) and Polymerase Chain Reaction (PCR), while a standardized form was used to collect sociodemographic, health, and behavioral data. Statistical analysis, conducted using Epi-Info 3.3.2, considered *T. gondii* seropositivity as the primary outcome, with a significance level less than 5% (*p* ≤ 0.05). An overall seroprevalence of 62.11% (100/161) was observed. Key risk factors significantly, as measured by the Odds Ratio (OR), associated with *T. gondii* seropositivity included being elderly (OR: 4.07, CI: 2.05–8.06, *p* < 0.01), having cats (OR: 5.56, CI: 2.74–22.27, *p* < 0.01), a low parental education level (OR: 2.97, CI: 1.46–6.02, *p* < 0.01), children playing on the ground (OR: 2.50, CI: 1.30–4.82, *p* < 0.01), and having a home vegetable garden (OR: 3.80, CI: 1.94–7.47, *p* < 0.01). Regarding ocular health, no conclusive direct association was established between *T. gondii* seropositivity and specific ocular manifestations when analyzed for children and the elderly separately. Observed ocular problems in the grouped population were primarily linked to age-related comorbidities rather than parasitic infection. High rates of *T. gondii* seropositivity, driven by specific environmental and socioeconomic factors, highlight the vulnerability of these communities, emphasizing the need for targeted preventive strategies.

## 1. Introduction

Toxoplasmosis, caused by the parasite *Toxoplasma gondii* [1], is a globally significant parasitic disease capable of infecting all vertebrate species [2,3]. Its transmission primarily occurs through the ingestion of contaminated food and water [1]. Key routes of human infection include consuming raw or undercooked meat containing tissue cysts, ingesting oocysts from contaminated soil (e.g., through unwashed vegetables or accidental contact), and vertical transmission [4,5,6]. Notably, parasitic contamination often correlates with conditions of social vulnerability, such as inadequate sanitary infrastructure, low income, and illiteracy [7,8]. Furthermore, populations in rural areas face heightened exposure risks due to vulnerable living conditions, diverse domestic and wild intermediate hosts, uncontrolled feline populations, and limited access to healthcare [2,6].

In Brazil, *Quilombola* communities are historically significant semi-isolated rural communities founded by descendants of enslaved African people, who maintained their cultural heritage and traditions through subsistence agriculture after the abolition of slavery in 1888 [9]. Officially recognized by the Brazilian Ministry of Health since 2004, these communities, totaling approximately 5972 across the country, are covered by inclusion policies aimed at expanding healthcare teams [10].

Despite the recognized exposure of *Quilombola* individuals to *T. gondii*, driven by an environment of multiple socioeconomic and environmental risk factors, the literature lacks studies evaluating these Brazilian populations. This gap represents a significant challenge for public health. Therefore, the present study aimed to assess the seroprevalence of anti-*T. gondii* Immunoglobulin Class G (IgG) and Immunoglobulin Class M (IgM) in *Quilombola* communities, to identify risk factors associated with the seropositivity for this infection, to isolate the parasite’s DNA using a polymerase chain reaction (PCR), and to investigate ocular health in this population, focusing on four communities in northern Brazil.

## 2. Materials and Methods

The reporting of this study adheres to the STROBE (Strengthening the Reporting of Observational Studies in Epidemiology) statement, which contains 22 items to be included in observational studies [11].

This safety study adheres to the principles of the Declaration of Helsinki for medical research involving human subjects, which includes the absolute priority of the well-being of the participants, the protection of their health and rights, respect for their autonomy and informed consent, rigorous assessment of the risks and benefits, and the obligation to discontinue research if the risks become unacceptable. It was approved by the Research Ethics Committee of the Tocantins University Center President Antônio Carlos—UNITPAC, under opinion number 3.070.168, approved on 10 December 2018.

The research was initiated only after authorization from the local leaders of each *Quilombolas* communities, who authorized the use of physical space for the research team and the research participants. The research was only carried out in the presence—in whole or in part—of the Family Health Program team responsible for the area.

Once the data collection was approved and the date confirmed, the health department of the corresponding region instructed the family health team responsible for each *Quilombolas* communities to pass on information related to the data collection process to individuals who met the requirements for inclusion in the research, requesting their attendance at the previously determined location, explaining the objectives of the examination, its benefits, the absence of onus on the participants, and the commitment to confidentiality regarding the diagnosis and conduct in each case. The inclusion of each participant took place after they signed the consent form to authorize his/her participation or the participation of the children under his/her responsibility.

A cross-sectional study was conducted in the period from March to September 2022 and involved 161 *Quilombolas* in the northern region of the state of Tocantins, Brazil, which is in the eastern region of the Brazilian Legal Amazon. These individuals lived in the *Quilombolas* communities certified by the Palmares Foundation, Brazil, in the period from March to September 2022 (Aragominas, Bavaria, Cocalinho, and Muricilândia). The *Quilombola* communities that presented the official certification were as follows: Pé do Morro and Bavaria, both in the municipality of Aragominas, Dona Juscelina, in the municipality of Muricilândia, and Cocalinho, in the municipality of Santa Fé. All the municipalities are located in the northern region of the state of Tocantins, Brazil (Figure 1).

All the inhabitants of the *Quilombola* communities, certified by the Palmares Foundation, who were part of the immediate geographical region of the municipality of Araguaína and who were aged between 5 and 7 years or equal to or over 60 years, were considered eligible for the study.

Children aged 5 to 7 years were included, since they are considered to be in the final period of visual maturation, and early intervention is crucial to prevent functional blindness. Elderly individuals over 60 years of age were also included, as they are in the main group affected by preventable blindness and low vision worldwide and classically present higher rates of toxoplasmic infection.

For data collection, protocols adapted from population studies were used, which involved sociodemographic variables (age, sex, race, address, place of birth, place of residence, education level, and occupation of parents) and epidemiological variables (presence of dogs or cats at home, contact with dogs or cats, contact with other animals; and lifestyle variables such as barefoot soil management, going to a beach or farm, contact with rivers or lakes, drinking untreated or unfiltered water, eating raw or undercooked meat, the place where foodstuffs—including meat, milk, eggs, fruits, and vegetables—were purchased, whether the residence had internal plumbing, the type of sewer system, and pre-existing illnesses).

The laboratory analysis to detect *T. gondii* infection was performed immediately after the visual acuity examination. From all the individuals who signed the informed consent form, 5 mL of peripheral blood was collected to perform the ELISA to screen for the presence of specific anti-*T. gondii* antibodies of the IgG and/or IgM classes. To preserve internal validity, the same serological test was used with the same brand of kit for all participants (Imunotoxo Bioclin-Quibasa S/A^®^, Quibasa Química Básica Ltda (Bioclin), Belo Horizonte, Minas Gerais, Brazil), and the analyses were performed in a single laboratory, thus avoiding measurement errors. The tests were centrally processed in the main laboratory of the municipality of Araguaína, Tocantins.

In cases where the serology was positive for IgM, polymerase chain reaction (PCR) was performed. The extraction of *T. gondii* DNA from the whole blood of *Quilombolas* was carried out using commercial DNA extraction kits, such as BIOPUR (Biometrix, Prides Crossing, MA, USA) and/or PureLink Genomic DNA Purification (Invitrogen, Carlsbad, CA, USA). The reaction was carried out in a final volume of 25 μL, containing 10 mM Tris-HCl (pH 9.0), 3.5 mM MgCl_2_, 0.2 U of Taq DNA Polymerase (Invitrogen), 0.5 mM of each deoxynucleotide (dATP, dTTP, dGTP, dCTP; Sigma Chemical Co., Burlington, MA, USA), 50 pmol of each primer (Invitrogen), and 2 μL of template DNA. To the reaction mixture, 40 μL of mineral oil (Sigma Chemical Co., USA) was added. The reactions were performed in a MasterCycler Personal thermocycler. The amplification program consisted of an initial denaturation at 94 °C (5 min), 35 denaturation cycles at 94 °C (1 min), annealing at 62 °C (1 min), and extension at 72 °C (1 min), followed by a final extension at 72 °C (10 min). The pairs of primers used were Toxo-B5 (5′-TGA AGA GAG GAA ACA GGT GGT CG-3′) and Toxo-B6 (5′-CCG CCT CCT TCG TCC GTC GTA-3′).

Previously tested positive and negative samples were used as controls. The PCR-amplified products, with a size of 100 bp, were visualized via electrophoresis on 6% polyacrylamide gel and silver staining.

An assessment of the population’s eye health was conducted in basic health units or schools in the *Quilombola* communities through a cross-sectional study to detect refractive errors and eye diseases. A structure with ophthalmological equipment was set up by the research team, and ophthalmologists were present to perform the examinations. Only the people who agreed to participate in the study were examined.

Ophthalmological acuity was determined based on the last line in which the participant correctly identified three or more optotypes. If the participant was unable to read the first optotype (first line), the vision was classified by the participants counting fingers, observing hand movement, citing light perception or no light perception, and in those cases in which the patient could not provide the correct information, the visual acuity was recorded as “not reported.”

The visual acuity was assessed, and the consensual and photomotor pupillary reflexes were evaluated to verify the appropriate pupillary response to light and the coordinated reaction of both eyes to light stimuli. Extrinsic eye movement was also assessed.

All the participants who presented with altered visual acuity underwent computerized refractometry—without cycloplegia in the case of elderly individuals and with cycloplegia in the case of children, achieved by putting one drop of 1% cyclopentolate into the lower conjunctival sac. Computerized refractometry was performed with an automatic refractor (KR 7000, Topcon, Tokyo, Japan), followed by dynamic subjective refraction with the interposition of corrective lenses until the best optical correction was found. The retina was evaluated under mydriasis using a binocular indirect ophthalmoscope (Eyetec OBI-LED, Brookfield, WI, USA) and a 20-diopter VOLK lens.

In patients who presented fundoscopic alterations on indirect binocular ophthalmoscopy, retinal images were captured with a portable retinal camera device (Eyer, Phelcon, São Carlos, Brazil), and the images were evaluated with the artificial intelligence software developed by the manufacturer.

The data collected were entered into a specific database generated in the Epi-Info 3.3.2 statistical program. This same program was used for statistical analysis. Initially, frequency distribution tables were constructed for the categorical variables. Subsequently, contingency tables were created to determine the association between the dependent variable (seropositivity for *T. gondii*, which is shown in the results of positive IgM and/or IgG and/or positive PCR laboratory tests) and the independent variables (sociodemographic, epidemiological, and eye health) using a multivariate logistic model, estimating an OR with a 95% confidence interval (CI95%) between the subgroups formed from each variable. The level of significance was set at 5% (*p*).

## 3. Results

From March to September 2022, 161 *Quilombolas*, 33 from Pé do Morro, 51 from Baviera, 28 from Dona Juscelina, and 49 from Cocalinho (81 children and 80 elderly people in total) were tested for anti-*T. gondii*, with 36/81 (44.44%) of children and 62/80 (77.50%) of elderly people testing positive for IgG. Two out of 81 (2.47%) children were found to have both IgG and IgM antibodies in their peripheral blood. The total number of children with circulating antibodies (IgG or IgG/IgM) was 38/81 (46.91%). No IgM antibodies were found in the peripheral blood of the elderly participants. Of the total number of subjects, 100/161 (62.11%) of the total sample were serologically positive, which was the primary outcome of the study. No cases with positive PCR tests were found (Table 1).

Regarding the sociodemographic and behavioral characteristics of the *Quilombola* children, it was observed that having cats, playing in the dirt, and having a garden were risk factors related to seropositivity for toxoplasmosis. No risk factors related to eye health problems were found in the group (Table 2).

Sociodemographic, epidemiological, and/or ocular health data for children not presented in relation to *T. gondii* infection are available in the Appendix A.

Regarding the sociodemographic and behavioral characteristics of elderly *Quilombolas*, it was observed that having cats, working in the soil, and having a garden were risk factors related to seropositivity for toxoplasmosis. No risk factors related to eye health problems were found in the group (Table 3).

Sociodemographic, epidemiological, and/or ocular health data for the elderly not presented in relation to *T. gondii* infection are available in the Appendix A.

Regarding the sociodemographic and behavioral characteristics of the grouped *Quilombola* children and elderly, it was observed that being a child was a protective factor against toxoplasmosis infection. Parents having low levels of education (less than 8 years), owning cats, working the land, having a garden, and having some pre-existing illness were risk factors related to seropositivity for toxoplasmosis (Table 4).

Changes in eye health, such as self-reported vision problems, cataracts, myopia, and blindness, were observed more frequently in the elderly than in children; however, these results had a similar prevalence to that of the general population, thus not characterizing a manifestation of *T. gondii* infection.

Sociodemographic, epidemiological, and/or ocular health data for the general population not presented in relation to *T. gondii* infection are available in the Appendix A.

## 4. Discussion

To our knowledge, this is the first study to evaluate the exposure to *T. gondii* among *Quilombolas*. This population has a life history marked by high socioeconomic vulnerability, low household income, low education, limited access to clean drinking water and healthcare services, and high levels of hunger and poverty [12].

The prevalence of seropositivity was 38/81 (46.91%) in children and 63/80 (77.78%) in the elderly, with an overall prevalence of 100/161 (62.11%). In Europe, the prevalence of toxoplasmosis in young people ranges from 13 to 16%, and in the elderly population, it is from 26 to 68% [13]. In Iran, in nomadic populations, the seroprevalence in children and young adults is 36%, and in the elderly population, it is 71.7% [14]. In Indigenous children in southern Brazil, in the state of Paraná, the prevalence was 24.9%, and in the elderly, it was 24% [15].

The results of this study are higher than the estimated combined seroprevalence of 36% for the general population worldwide, as determined by a global meta-analysis [16]. In Pakistan, the reported prevalence was 33.5% [17]; in Mexico, it was 32.9% [18]; in São Paulo, southeastern Brazil, it was 48% [19]; in Tunisia, it was 25.3% [20]; while in Peru, the prevalence was 83.3% in an Indigenous village subsisting on hunting [21].

Our findings reveal an exceptionally high seroprevalence of *T. gondii* among children in these *Quilombola* communities, markedly surpassing the national and global averages. This result strongly indicates that, relative to other populations worldwide, children in this study are likely acquiring the infection at a very early age. This early exposure is a critical public health concern, underscoring the urgent necessity for implementing robust and culturally sensitive preventive interventions.

Such measures must include targeted educational programs and practical guidance on hygiene and environmental management, specifically designed for early childhood. Addressing this childhood vulnerability is paramount to mitigate the long-term health burden of toxoplasmosis, particularly given the socioeconomic and environmental vulnerabilities prevalent in these communities.

The seroprevalence in the elderly participants was similar to the findings of studies in other parts of the world, which also showed high values. This is consistent with the literature, since the elderly population has had more time to be exposed to the risk factors for infection [21,22,23,24]. One point of concern is that exposure to the parasite has been associated with poor vision or complete blindness [25].

Pre-existing health conditions refer to any chronic health conditions that an individual may have, such as immunocompromisation (HIV/AIDS, organ transplantation, or malignant neoplasms), which may promote toxoplasmic reactivation. In the pooled analysis, we found a significant association between the presence of “some disease” and seropositivity for toxoplasmosis. However, this result should be interpreted with caution, since in the separate analysis between seropositive and seronegative elderly individuals, no relationship was found, showing that this factor was not significant (Appendix A). This result differs from what was expected, since a correlation between poor health and acute clinical manifestation of toxoplasmic infection is often found.

The sociodemographic and epidemiological factors associated with seropositivity for *T. gondii* were similar in the child and elderly participants. These included a parental education level of less than 8 years of schooling, having a yard with soil, owning cats, handling soil, frequenting rural properties or farms, and having a garden. These findings corroborate the fact that the environment, behavior, and the presence of cats directly influence the increase in the seroprevalence rates of this infection [26,27,28]. Low parental education levels influence childhood infection by limiting the awareness of *T. gondii* transmission routes and preventive practices. This can result in poorer adherence to hygiene habits and proper environmental management, increasing children’s exposure to risk factors in the home environment.

Cats, who are the definitive hosts of this infection and contaminate the environment with millions of oocysts in the soil, create an unsanitary and extremely risky environment [22,23,29,30].

Felines become infected through carnivory, and sexual reproduction occurs in the epithelium of their gastrointestinal tract, making them the definitive host. Through the excrement of these animals, *T. gondii* contaminates the soil with oocysts, which can remain viable for up to 18 months, especially in moist soils in regions with a temperate climate [31].

In cats, during infection, millions of oocysts are excreted in their feces over a period of 7 to 21 days. Humans and animals become infected by ingesting oocysts in raw foods that have come into contact with soil; through activities involving direct handling of soil; by drinking or coming into contact with contaminated water; or through direct contact with infected cats [4,31,32,33]. Gardens pose a high risk, because cats, definitive hosts of T. gondii, shed viable oocysts for long periods in their feces in moist soil. Furthermore, the data indicate that working on land not near the home was a protective factor against *Toxoplasma* infection. Considering that cats are territorial animals by nature, when they live in rural areas or have access to outdoor areas, it is common for them to urinate or defecate in soil or yard areas, which is associated with both their territorial marking and excretory habits. Thus, children and the elderly who spend most of the day in the vicinity of their homes are more likely to become infected (see Appendix A). However, it seems that the crucial factor in this relationship is the lack of education of the parents who, unaware of the presence of this risk, do not protect themselves or their families.

Despite the classical association of *T. gondii* with severe ocular damage, such as chorioretinitis [34,35,36,37,38,39,40,41,42], and the observation of ocular problems being more present in the infected group when the population (children and elderly) was analyzed in a grouped manner, our study did not establish a direct and conclusive association between seropositivity and specific ocular manifestations when analyzing the age groups separately.

The assessment of ocular manifestations solely by direct ophthalmoscopy in individuals with altered visual examinations, coupled with the absence of perimetry, may represent a limitation in our study. It is crucial to recognize that visual acuity alone can underestimate functional damage, as peripheral chorioretinal lesions, which are often difficult to detect by restricted direct ophthalmoscopy, can severely impact the visual field. This distinction is particularly important, as visual field defects are significantly more prevalent (94%) than reduced visual acuity (41%) in cases of inactive ocular toxoplasmosis, indicating that visual acuity alone may underestimate functional damage, which could lead to a potential underestimation of the true extent of ocular involvement in this population [43].

This segregated approach was crucial to avoid misinterpretation of the results, as aging and its associated visual comorbidities are sufficient causes for ocular lesions and impairments. The absence of significant results when comparing exposed and unexposed elderly individuals (see Appendix A) reinforces the importance of this distinction.

The absence of a statistically significant association between toxoplasmosis and ocular damage is likely due to the limited sample size. This limitation may have increased the risk of a Type II error and should be considered when interpreting the results.

Future longitudinal studies, with larger samples and follow-up on the progression of potential lesions, would be valuable to further understand the relationship between *T. gondii* infection and ocular health in these vulnerable populations, ensuring that any parasitic impact is distinctly assessed separately from age-related comorbidities.

## 5. Conclusions

Our study uniquely revealed a remarkably high prevalence of *T. gondii* seropositivity among the *Quilombola* population of northern Tocantins, Brazil; this was particularly pronounced in children, where the rates significantly exceeded the global estimates. This critical finding demands urgent attention and underscores the need for targeted preventive actions starting in childhood.

This research identified a robust set of interconnected risk factors directly linked to the unique socioeconomic and environmental conditions prevalent in these historically marginalized communities, such *Quilombola* communities as advanced age, cat contact, low parental education, regular contact with soil, and the presence of home gardens. Collectively, these factors highlight a complex interaction between environmental exposure, behavioral patterns, and educational deficiencies that lead to high infection rates, emphasizing the profound impact of social and environmental vulnerability on health outcomes.

As for eye health, our study meticulously explored its relationship with *T. gondii* infection. Although the parasite is classically associated with eye damage, such as chorioretinitis, this study was unable to establish a direct and conclusive association between seropositivity and ocular manifestations. The observed increase in eye problems (myopia, cataracts, blindness) in the grouped population was attributable to age-related comorbidities rather than a direct consequence of parasitic infection. This nuance is crucial to avoid misinterpretation of the results and highlights the importance of contextual analysis.

Ultimately, this study fills a significant gap in knowledge about the epidemiology of toxoplasmosis in Brazil. Our results provide a robust basis for the development of targeted public health interventions and educational programs aimed at reducing the burden of *T. gondii* infection and promoting better health outcomes in these historically underserved groups.

## Figures and Tables

**Figure 1 microorganisms-14-00096-f001:**
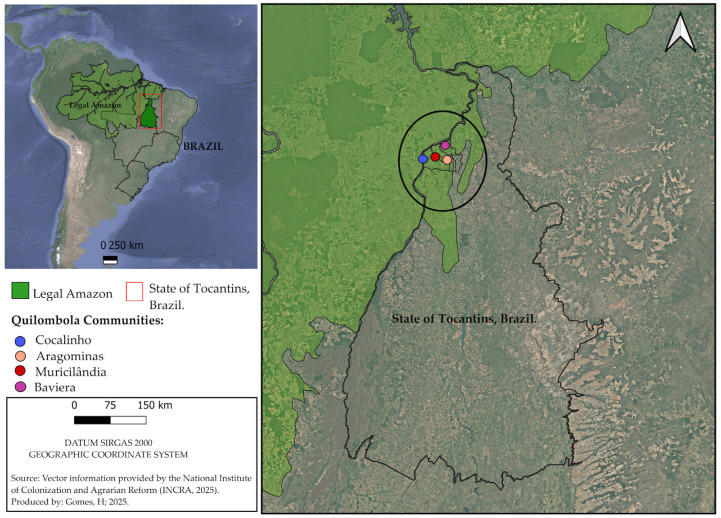
Geographic location of the *Quilombola* communities that formed the study group, located in the eastern region of the Brazilian Legal Amazon, in the state of Tocantins, Brazil.

**Table 1 microorganisms-14-00096-t001:** Laboratory results (IgG, IgG and IgM, IgM, and PCR serology) for *Toxoplasma gondii* in *Quilombolas*.

Test	Children		Elderly		Total				
	*n*	%	*n*	%	*n*	%	OR	CI 95%	*p*
IgG	36/81	44.44	62/80	77.50	98/161	60.87	4.31	2.17–8.53	<0.0001
IgG and IgM	2/81	2.47%	-	-	2/161	1.24	-	-	-
IgM	-	-	-	-	-	-	-	-	-
PCR	-	-	-	-	-	-	-	-	-
Total	38/81	46.91	62/80	77.50	100/161	62.11	3.90	1.97–7.71	0.0001

*n*: number, %: percentage, OR: odds ratio, CI: confidence interval, *p*: significance level.

**Table 2 microorganisms-14-00096-t002:** Sociodemographic characteristics and analysis of epidemiological and ocular characteristics related to *Toxoplasma gondii* seropositivity in the *Quilombola* children.

		*n* (Positive)/Total Children		*n* (Negative)/Total Children		OR	CI 95%	*p*
		n	%	n	%			
Has cats	Yes	23	60.53	10	23.25	5.06	1.93–13.23	0.001
	No	15	39.47	33	76.74			
Has a garden	Yes	25	65.79	15	34.88	3.59	1.43–8.99	0.005
	No	13	34.21	28	65.12			
Plays in the soil	Yes	35	92.11	21	95.35	12.22	3.26–45.85	0.0001
	No	3	7.89	22	4.65			

*n*: number, %: percentage, OR: odds ratio, CI: confidence interval, *p*: significance level.

**Table 3 microorganisms-14-00096-t003:** Sociodemographic characteristics and analysis of epidemiological and ocular characteristics related to *Toxoplasma gondii* seropositivity in elderly *Quilombolas*.

		*n* (Positive)/Total Elderly		*n* (Negative)/Total Elderly		OR	CI 95%	*p*
		n	%	n	%			
Has a garden	Yes	40	64.52	5	50.00	4.73	1.49–15.00	0.013
	No	22	35.48	13	50.00			
Has a yard with soil	Yes	57	91.94	11	61.11	7.25	1.94–27.08	0.004
	No	5	8.06	7	38.89			
Has cats	Yes	40	64.52	5	27.78	4.72	1.49–15.01	0.013
	No	22	35.48	13	72.22			

*n*: number, %: percentage, OR: odds ratio, CI: confidence interval, *p*: significance level.

**Table 4 microorganisms-14-00096-t004:** Sociodemographic characteristics and analysis of epidemiological and ocular characteristics related to seropositivity for *Toxoplasma gondii* in children and elderly individuals from *Quilombola* communities.

		*n* (Positive)/Total		*n* (Negative)/Total		OR	CI 95%	*p*
		n	%	n	%			
Age	≥5 ≤ 7	38	38.00	43	70.49	4.07	2.05–8.06	0.00004
	≥60	62	62.00	18	29.51			
Distance visual acuity	20/20	34	34.00	31	50.82	-	-	-
	20/30–20/60	21	21.00	19	31.15	1.01	0.46–2.22	0.458
	20/70–20/160	20	20.00	7	11.48	2.60	0.97–7.00	0.089
	20/200–20/400	8	8.00	2	3.28	3.65	0.72–15.50	0.193
	Blindness	12	12.00	1	1.64	10.94	1.34–89.10	0.018
	Without light perception	4	4.00	0	0.00	-	-	-
	Luminous perception	0	0.00	1	1.64	-	-	-
Goes to a farm, ranch, or poultry farm	Yes	51	51.00	41	67.21	0.51	0.26–0.98	0.044
	No	49	49.00	20	32.79			
Has any illness	Yes	38	38.00	10	16.39	3.13	1.42–6.88	0.004
	No	62	62.00	51	83.61			
Has noticed any vision problems	Yes	63	63.00	23	37.70	2.81	1.46–5.43	0.002
	No	37	37.00	38	62.30			
Lens	Opaque	37	39.78	10	16.67	3.30	1.49–7.32	0.002
	Normal	56	60.22	50	83.33			
Parents’ education level	≤8 years	80	80.00	35	52.46	2.97	1.46–6.02	0.00004
	>8 years	20	20.00	26	47.54			
Spherical equivalent	Normal vision	42	42.00	35	57.38	-	-	-
	Myopia	31	31.00	8	13.11	3.23	1.32–7.92	0.015
	Hyperopia	27	27.00	18	29.51	1.25	0.59–2.64	0.691

*n*: number, %: percentage, OR: odds ratio, CI: confidence interval, *p*: significance level.

## Data Availability

The data used in this study are available in an Excel 2021 spreadsheet at the following link: https://docs.google.com/spreadsheets/d/14AADSCvowb9tdYn9lVxHqzTCgmZElmQu/edit?usp=sharing&ouid=104283245732475613436&rtpof=true&sd=true, accessed on 29 December 2025.

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
