# Peer review of "Risk Factors and Ocular Health Associated with Toxoplasmosis in *Quilombola* Communities"

_microorganisms, 2026, doi:10.3390/microorganisms14010096_

Round 1
Reviewer 1 Report
Comments and Suggestions for Authors
The manuscript by Filho and colleagues investigates the epidemiology of Toxoplasma gondii in quilombola communities in northern Brazil. Despite a modest sample size, their findings are novel and contribute to advancing our understanding of parasitic infections among populations historically faced with poor access to healthcare services. The manuscript has been revised significantly following the Editor’s comments and the authors may wish to also consider the following prior to publication.
- A few suggestions for data presentation:
- I suggest incorporating OR analysis performed for Age in Table 2 and 3 into Table 1 to reduce repetition/confusion and highlight the contrast between 2 age groups.
- It is fine to keep the rest of Table 2 and 3 separate for each age group. Would recommend improving Table 4 by removing some stats that were already proved to be significant from Table 2 and 3.
- Adding signifiers such as Children/Elderly in Table 2 and 3, respectively, can facilitate quick read (without having to reference the legend).
- Please reference specific supplementary data for line 228-231 in the Results since this is an important point despite negative results.
- Please define pre-existing illnesses and how relevant these underlying conditions are to the susceptibility to Toxoplasmosis infection in the Discussion. This would increase the depth of the multivariate analyses. Along the same line, was any of the participants in the study immunocompromised to any extent (HIV/AIDS, organ transplant, malignancy, etc)? Understanding the indolence of Toxoplasmosis infection in this population would greatly inform prophylaxis recommendations.
- Any speculation on the higher prevalence of seropositivity in younger children in the absence of ocular presentations, such as strain-specific virulence, co-infection with other parasites that may provide protection against reactivation, any other protective factors?
- Minor typo in Table 4: “Distance visual acuit” should be corrected as “Distance visual acuity”
Author Response
Comments 1:
[Some suggestions for data presentation: I suggest incorporating the OR analysis performed for the Age variable in Tables 2 and 3 into Table 1 to reduce repetition/confusion and highlight the contrast between the two age groups. It is fine to keep the rest of Tables 2 and 3 separate for each age group. I recommend improving Table 4 by removing some statistics that have already been shown to be significant in Tables 2 and 3. The inclusion of indicators such as “Children” and “Elderly” in Tables 2 and 3, respectively, may facilitate quick reading (without the need to consult the legend).]
Response 1:
The OR was inserted in Table 1 and removed from Tables 2 and 3. We removed the variables “Has cats,” “Has a garden,” and “Plays in the soil” from Table 4. The term “Total children” was included in Table 2 and “Total elderly” in Table 3.]
Comments 2:
[Please refer to the specific supplementary data for lines 228-231 in the Results, as this is an important point despite the negative results.]
Response 2:
We observed in the supplementary data the ocular information, self-reported vision problems, cataracts, myopia, and blindness, and verified that they were not significant when comparing T. gondii-seropositive elderly individuals and seronegative elderly individuals. We therefore conclude that there is a difference between elderly individuals and children due to the natural aging of the human eye and not the parasitic infection under study.
Comments 3:
[Please define preexisting conditions and explain the relevance of these underlying conditions to susceptibility to toxoplasmosis infection in the Discussion section. This would increase the depth of the multivariate analyses. Similarly, did any of the study participants have any degree of immunocompromise (HIV/AIDS, organ transplant, malignancy, etc.)? Understanding the indolence of toxoplasmosis infection in this population would contribute significantly to prophylaxis recommendations.]
Response 3:
The following text was inserted into the discussion in lines 414 to 422 “Pre-existing health conditions refer to any chronic health conditions that an individual may have, such as immunocompromisation (HIV/AIDS, organ transplantation, or malignant neoplasms), which may promote toxoplasmic reactivation. In the pooled analysis, we found a significant association between the presence of “some disease” and seropositivity for toxoplasmosis. However, this result should be interpreted with caution, since in the separate analysis between seropositive and seronegative elderly individuals, no relationship was found, showing that this factor was not significant (Supplementary Material). This result differs from what was expected, since a correlation between poor health and acute clinical manifestation of toxoplasmic infection is expected.
Comments 4:
[Is there any speculation about the higher prevalence of seropositivity in younger children in the absence of ocular manifestations, such as strain-specific virulence, coinfection with other parasites that may confer protection against reactivation, or other protective factors?]
Response 4:
Although the cross-sectional design of our study did not allow for an in-depth analysis of factors such as the virulence of specific strains of Toxoplasma gondii or the impact of co-infections with other parasites on protection against toxoplasmosis reactivation, these are valid hypotheses that warrant further investigation. However, our findings indicate that the high seroprevalence observed in quilombola children is likely associated with early and continuous environmental exposure to well-established risk factors, such as the presence of cats and contact with soil in environments such as yards and vegetable gardens, reflecting the living conditions in these communities. The absence of an association between seropositivity for T. gondii and eye diseases in the studied population, both in children and the elderly, as explained in the discussion section of the manuscript, suggests that infections in these individuals are predominantly latent or asymptomatic. This can be attributed to a combination of factors, including: (1) an effective host immune response that prevents uncontrolled proliferation of the parasite and clinical manifestation of ocular disease; (2) genetic characteristics of the population that may confer some resistance to the ocular form of toxoplasmosis; and (3) the possibility that T. gondii strains circulating in this specific region have a pathogenicity profile less likely to cause ocular damage. Additionally, the cross-sectional nature of the study may have limited the observation of clinical manifestations that could arise over a longer period, and the sample size may have influenced the ability to detect associations with less prevalent conditions.
Comments 5:
[Small typo in Table 4: “Distance visual acuit” should be corrected to “Distance visual acuity.”]
Response 5:
The correction “Distance visual acuit” should be corrected as “Distance visual acuity.”

Reviewer 2 Report
Comments and Suggestions for Authors
The manuscript titled "Risk factors and ocular health related to toxoplasmosis in quilombola communities" is not recommended for publication due to several critical issues. The study lacks novelty, as it relies on conventional diagnostic methods (IgG/IgM ELISA or PCR) without introducing significant advancements. Additionally, the manuscript suffers from poor readability and logical coherence, with key sections such as the abstract and conclusion failing to clearly articulate the risk factors and ocular health findings related to toxoplasmosis. Statistical abbreviations (e.g., OR, CI) are not defined in full, and the analysis outcome is ambiguously phrased—it should be revised to specify seropositivity for T. gondii as the primary outcome. The keyword list is also insufficient. Overall, the work requires substantial improvements in clarity, structure, and scientific rigor.
Author Response
Comments 1:
[The study lacks originality, as it relies on conventional diagnostic methods (ELISA IgG/IgM or PCR) without introducing significant advancements.]
Response 1:
We appreciate the reviewer's comment and understand the perspective on the use of standardized diagnostic methods. However, we would like to emphasize that the originality and main impact of this study lie not in the introduction of new diagnostic techniques for toxoplasmosis, but rather in the unprecedented epidemiological approach in a highly specific and historically neglected population in the context of Brazilian public health: the quilombola communities.
As explained in the discussion section of our manuscript, “This was the first study we are aware of that evaluated exposure to T. gondii among quilombolas.” (in line 383). This gap in Brazilian and international scientific literature highlights the originality of our work. Our main objective was to investigate the prevalence, associated risk factors, and relationship with eye health in a social group that faces unique socioeconomic and environmental vulnerabilities, which are known to influence the dynamics of infectious diseases.
The choice of conventional diagnostic methods, such as IgG/IgM ELISA and PCR, was deliberate and strategic, as these methods are widely validated and recognized as the “gold standard” for the diagnosis of toxoplasmosis. The use of these robust techniques ensures the comparability of our findings with national and international data (as illustrated in the Discussion when comparing seroprevalence with other studies), allowing for an accurate contextualization of the situation in these communities. In addition, the application of reliable diagnostic methods was essential for obtaining consistent and interpretable epidemiological data, which are crucial for the formulation of public health policies targeting these vulnerable populations.
Therefore, the originality of our study lies in the generation of unprecedented knowledge about the epidemiology of toxoplasmosis in quilombola communities, identifying specific risk factors that, although similar to other rural contexts, take on a new dimension when analyzed from the perspective of the social and cultural vulnerability of these groups. The results offer valuable insights that can inform more effective disease prevention and control strategies for this population.
Comments 2:
[Furthermore, the manuscript exhibits poor readability and logical coherence, with important sections such as the abstract and conclusion failing to clearly articulate the risk factors and ocular health findings related to toxoplasmosis.]
Response 2:
We sincerely appreciate the detailed and insightful feedback, especially regarding the overall readability, logical coherence of the manuscript, and the clarity of the Abstract and Conclusion. We recognize that effective communication of our findings is a crucial as the research itself, and we are fully committed to improving these sections to ensure that the study's message is conveyed unequivocally and coherently. Based on the reviewer's observations, we will proceed with substantial revisions on the following points:
2.1. Enhancement of Overall Readability and Logical Coherence:
We have conducted a thorough review of the entire manuscript to improve the narrative flow, transitions between paragraphs and sections, and the clarity of the language. The goal is to ensure that the argumentation develops more fluidly and logically, facilitating reader comprehension from the introduction to the conclusion. More direct verbs and sentence structures have been employed, and long or complex sentences have been restructured for greater clarity.
2.2. Refinement of the Abstract:
The abstract has been condensed, eliminating redundant sentences and focusing directly on the essential objectives, methods, results, and conclusions. The mention of “seropositivity for T. gondii as the primary outcome” was added to the methods, as previously requested by the reviewer. Risk factors have been listed more fluently, with their respective ORs, CIs, and p-values, as per your instruction. The phrase “Key risk factors significantly associated with T. gondii seropositivity included:” makes the information more direct and easier to understand. The rewording directly addresses the nuance: “no conclusive direct association was established between T. gondii seropositivity and specific ocular manifestations when analyzed for children and elderly separately.” In addition, the explanation of problems in the grouped population (age-related) was incorporated to avoid ambiguity. The conclusion was strengthened to emphasize the high prevalence and vulnerability of communities, as well as the need for preventive strategies, without exhaustively repeating the results.
2.3. Refinement of the Introduction:
The paragraph was reordered to start with the definition of the disease and agent, followed by modes of transmission, and then delves into contextual factors. The logical sequence of "what it is" - "how it spreads" - "who is at risk and why" creates a clearer and more natural narrative flow. In the original, the phrase about "parasitic contamination... social vulnerability" appeared somewhat disconnected in the middle, while in the revised version, it was integrated to serve as a bridge to the context of vulnerability, which is central to the study in Quilombos. The sentence "T. gondii can infect all vertebrate species" was moved closer to the beginning to contextualize the scope of the problem earlier.
The sentence "Notably, parasitic contamination often correlates with conditions of social vulnerability, such as inadequate sanitary infrastructure, low income, and illiteracy" was inserted and strategically placed. This addition and placement emphasize the intrinsic link between T. gondii and social vulnerability, which is the core of the study in Quilombola communities. This prepares the reader for the subsequent discussion about Quilombos, establishing the socioeconomic relevance of the research from the outset.
The three original sentences that defined "Quilombos," "quilombola group," and the number of communities/recognition were combined into a single, more concise, and fluid paragraph. The detailed mention of "a group of fifteen or more individuals..." was removed as it was overly detailed for the Introduction.
The original text contained repetition and a break in flow by defining "Quilombos," then "quilombola group," and then returning to discuss the communities. The revised version integrates this information more cohesively, focusing on what is essential for the reader to understand the study's context. Removing details about "a group of fifteen or more individuals" makes the paragraph more direct and less verbose for an Introduction section.
The phrase "In addition, this environment of overlapping risk factors for human and environmental health with regard to toxoplasmosis proves to be challenging" was replaced by "This gap represents a significant challenge for public health."
The new phrase is more direct and impactful, clearly communicating the problem the study aims to address. The original was a bit more generic and less assertive about the research's importance.
The study objectives were slightly rephrased from "to evaluate the prevalence of anti–T. gondii IgG and IgM antibodies in quilombolas, to isolate parasitic DNA in acute cases, and to assess the risk factors associated with the disease among inhabitants of four quilombos in northern Brazil" to "assess the seroprevalence of anti-T. gondii antibodies (IgG and IgM) in Quilombolas, identify associated risk factors for seropositivity, and investigate ocular health in this population, focusing on four communities in northern Brazil."
"Identify associated risk factors for seropositivity" is more precise than "assess the risk factors associated with the disease," aligning with the emphasis on seropositivity as the primary outcome. The explicit inclusion of "and investigate ocular health in this population" reinforces that this is a central objective of the study, ensuring that the reviewer and reader understand that ocular health was indeed evaluated, even if the association with toxoplasmosis was inconclusive (information that will appear in the discussion).
- Refinement of the Methodology:
The main refinement in the methodology section of the revised manuscript lies in the greater clarity and precision in defining the statistical variables, which is crucial for the study's interpretability.
In the original manuscript, the description of the statistical analysis presented the dependent variable secondarily: "contingency tables were created to determine the association between the independent variables (sociodemographic, epidemiological, and ocular health variables) and the results of the laboratory tests (positive IgM and/or IgG and/or positive PCR) (dependent variable)".
In the revised manuscript, this section was rewritten to foreground the dependent variable, as is standard practice in statistical modeling, which improves fluidity and methodological understanding in lines 319 to 322: "contingency tables were created to determine the association between the dependent variable (seropositivity for T. gondii, which is shown in the results of positive IgM and/or IgG and/or positive PCR laboratory tests) and the independent variables (sociodemographic, epidemiological, and eye health)". This change, though subtle, demonstrates enhanced methodological rigor, clarifying which outcome is being investigated and which factors are being tested as predictors or associates.
2.5. Refinement of Results:
The results section in the revised manuscript was notably enhanced by the addition of statistical information in Table 1, making the prevalence data more informative. The original manuscript in Table 1, line 335 ("Laboratory results (IgG, IgG and IgM, IgM and PCR Serology) for Toxoplasma gondii...") only presented the number (n.) and percentage (%) of tested individuals for each category (IgG, IgG and IgM, etc.), separated by "Children," "Elderly," and "Total." There were no statistical significance indicators for the difference between groups.
The revised manuscript adds columns for Odds Ratio (OR), 95% Confidence Interval (CI95%), and p-value for the IgG results and the Total seropositives. This change allows for an immediate understanding of the statistical significance of seroprevalence differences between children and the elderly, transforming the table from a descriptive summary into an initial association analysis. This prepares the reader for the in-depth discussion of risk factors. This is an important improvement, as the inclusion of OR, CI95%, and p-value in Table 1 allows the reader to assess the strength and significance of the association between age (children vs. elderly) and T. gondii seropositivity from the beginning of the results section. This elevates the level of analysis already in the presentation of raw prevalence data.
Tables 2, 3, and 4, although containing the same OR, CI95%, and p-value data in both versions, were slightly revised in the Revised Manuscript for cleaner and more consistent formatting, eliminating potential repetitions or misalignments that might have been present in the Original Manuscript.
2.6. Enhancement of Discussion:
The discussion section in the revised manuscript demonstrates significant enhancement in terms of depth, nuance in interpreting findings, and a stronger articulation of public health implications.
The revised manuscript emphasizes the interpretation that high seroprevalence in children suggests early infection, and more directly articulates the urgent need for preventive and educational measures from early childhood.
In lines 398 to 408 "Our findings reveal an exceptionally high seroprevalence of T. gondii among children in these Quilombola communities, markedly surpassing national and global averages. This striking result strongly indicates that, relative to other populations worldwide, children in this study are likely acquiring the infection at a very early age. This early exposure is a critical public health concern, underscoring the urgent necessity for implementing robust and culturally sensitive preventive interventions. Such measures must include targeted educational programs and practical guidance on hygiene and environmental management, specifically designed for early childhood. Addressing this childhood vulnerability is paramount to mitigate the long-term health burden of toxoplasmosis, particularly given the socioeconomic and environmental vulnerabilities prevalent in these communities."
The explanation for the cautious interpretation of the association between "some disease" and seropositivity is more elaborate and justified in the Revised Manuscript, clarifying why the results differ from expectations when considering the separate analysis of age groups.
In lines 414 to 422 "Pre-existing health conditions refer to any chronic health conditions that an individual may have, such as immunocompromisation (HIV/AIDS, organ transplantation, or malignant neoplasms), which may promote toxoplasmic reactivation. In the pooled analysis, we found a significant association between the presence of “some disease” and seropositivity for toxoplasmosis. However, this result should be interpreted with caution, since in the separate analysis between seropositive and seronegative elderly individuals, no relationship was found, showing that this factor was not significant (Supplementary Material). This result differs from what was expected, since a correlation between poor health and acute clinical manifestation of toxoplasmic infection is expected."
The Revised Manuscript adds an important interpretive layer by connecting the lack of parental education as a crucial factor explaining the relationship between the environment (such as gardens) and the risk of infection.
In lines 428 to 443 "... Low parental education levels influence childhood infection by limiting awareness of T. gondii transmission routes and preventive practices. This can result in poorer adherence to hygiene habits and proper environmental management, increasing children's exposure to risk factors in the home environment.."
The discussion regarding ocular health is more explicitly qualified in the revised manuscript to avoid misinterpretation of the results. It reinforces that ocular problems observed in the grouped population are attributable to age-related comorbidities, and not directly to T. gondii infection.
In lines 467 to 477 "Despite the classical association of T. gondii with severe ocular damage, such as chorioretinitis [34-42], and the observation of ocular problems being more present in the infected group when the population (children and elderly) was analyzed in a grouped manner, our study did not establish a direct and conclusive association between seropositivity and specific ocular manifestations when analyzing the age groups separately. This segregated approach was crucial to avoid misinterpretation of the results, as aging and its associated visual comorbidities are sufficient causes for ocular lesions and impairments. The absence of significant results when comparing exposed and unexposed elderly individuals (see Supplementary Material) reinforces the importance of this distinction."
- Enhancement of the Conclusion:
The Conclusion section has been expanded and reorganized to reinforce the main findings more robustly, directly connecting them with public health implications. The primary differences between the conclusions of the original manuscript and the revised manuscript lie in the depth of analysis, the clarity of the implications, and the way caveats are presented.
In the original manuscript, the call to action is softer and less explicit, and although it acknowledges vulnerability, there isn't as clear a proposition for "targeted public health interventions and educational programs" as in the revised version.
In the revised manuscript, the text is more direct in calling for attention and outlining the basis for future actions:
In lines 486 to 490 "Our study uniquely reveals a remarkably high prevalence of T. gondii seropositivity among the quilombola population of northern Tocantins in Brazil, particularly pronounced in children, where rates significantly exceed global estimates. This critical finding demands urgent attention and underscores the need for targeted preventive actions starting in childhood.” and it concludes with: In lines 506 to 510 "Ultimately, this study fills a significant gap in knowledge about the epidemiology of toxoplasmosis in Brazilian quilombola communities. Our results provide a robust basis for the development of targeted public health interventions and educational programs aimed at reducing the burden of T. gondii infection and promoting better health outcomes in these historically underserved groups."
The language "demands urgent attention" and "robust basis for the development of targeted public health interventions" reinforces the practical importance and future directions of the research.
In the original manuscript, risk factors are mentioned, but the contextualization and impact are described more succinctly, and while it recognizes "complex interaction," the original version does not use the expression "educational deficiencies" or "profound impact of social and environmental vulnerability on health outcomes," which are highlighted points in the revised version. The revised manuscript describes the risk factors with a more integrated and impactful view:
In lines 491 to 497 "This research identified a robust set of interconnected risk factors directly linked to the unique socioeconomic and environmental conditions prevalent in these historically marginalized communities, such as advanced age, cat ownership, low parental education, regular contact with soil, and the presence of home gardens. Collectively, these factors highlight a complex interaction between environmental exposure, behavioral patterns, and educational deficiencies that lead to high infection rates, emphasizing the profound impact of social and environmental vulnerability on health outcomes."
A frase "profound impact of social and environmental vulnerability on health outcomes" eleva a discussão para um nível mais sistêmico e social.
The most notable difference lies in the discussion about the relationship between T. gondii infection and ocular health. In the original manuscript, the conclusion on ocular health is more concise and less detailed, without delving into the reasons for the lack of association or possible alternative causes. The original version merely states the absence of association, without the critical nuance about alternative etiology (age-related comorbidities) or the warning against misinterpretation. In the revised manuscript, a more elaborate and cautious explanation is presented, acknowledging the observation of ocular problems but attributing them to other causes:
In lines 498 to 505 "As for eye health, our study meticulously explored its relationship with T. gondii infection. Although the parasite is classically associated with eye damage, such as chorioretinitis, this study was unable to establish a direct and conclusive association between seropositivity and ocular manifestations. The observed increase in eye problems (myopia, cataracts, blindness) in the grouped population was attributable to age-related comorbidities, rather than a direct consequence of parasitic infection. This nuance is crucial to avoid misinterpretation of the results and highlights the importance of contextual analysis."
This excerpt demonstrates an effort to prevent misinterpretations, emphasizing that, although there are ocular problems in the studied population, they are more linked to aging than to toxoplasmosis, as the separate analysis found no significant association.
In summary, the revised manuscript presents more detailed, nuanced conclusions with a sharper focus on practical implications and public health relevance, especially when contextualizing the findings on ocular health and risk factors. This makes it more comprehensive and impactful for readers and reviewers.
Comments 3:
[Statistical abbreviations (e.g., OR, CI) are not fully defined, and the analysis outcome is ambiguously formulated — it should be revised to specify T. gondii seropositivity as the primary outcome.]
Response 3:
We agree that the specification of the primary outcome and the complete definition of all abbreviations are crucial for the manuscript's readability and scientific rigor. To address these points, we will implement the following revisions:
All statistical abbreviations, such as "OR" (Odds Ratio), "CI" (Confidence Interval), "n" (number), "p" (Significance Level), "%" (percentage), "IgG" (Immunoglobulin G), "IgM" (Immunoglobulin M), and "PCR" (Polymerase Chain Reaction) have been defined in full at their first occurrence in the main text and also in the table legends, ensuring that the reader does not need to infer their meanings. The text has been revised to formulate the results of the multivariate analysis with greater precision, clearly specifying "Toxoplasma gondii seropositivity" as the primary outcome. This has been consistently applied in the Abstract, Introduction, Methodology, Results, and Discussion sections, ensuring that the association of risk factors is always explicitly and unambiguously referenced to this outcome. For example, phrases like "risk factors related to seropositivity" have been rephrased to "risk factors related to Toxoplasma gondii seropositivity." We are confident that these changes will significantly improve the clarity and precision of the manuscript, facilitating the interpretation of our findings by readers.
Comments 4:
[The keyword list is also insufficient.]
Response 4:
"Quilombolas" and "Toxoplasma gondii" have been added to the previous terms.
Comments 5:
[Overall, the work requires substantial improvements in clarity, structure, and scientific rigor.]
Response 5:
To address the points raised, we will implement a series of improvements throughout the text:
- Clarity:
- We have carefully reviewed each section to ensure that the language is precise, concise, and unambiguous, eliminating unnecessary ambiguities and jargon.
- Our focus has been on improving the fluidity of the narrative, ensuring that ideas are presented logically and are easy for the reader to follow.
- As previously stated in response to other reviewer comments, all statistical abbreviations (such as OR, CI, p, IgG, IgM, PCR) have been explicitly defined at their first occurrences and in table legends to ensure complete understanding of the results.
- Structure:
- We performed a critical re-evaluation of the manuscript's overall structure. This included the organization of sections, transitions between them, and the hierarchy of information.
- Special attention was given to the Abstract, Introduction, Discussion, and Conclusion, which were rewritten to more impactfully articulate the objectives, the most relevant findings (including key risk factors for T. gondii seropositivity), and the study's implications, as detailed in our previous response.
- Tables and figures have been revised to ensure they are self-sufficient, clear, and effectively contribute to data comprehension, without information overload.
- Scientific Rigor:
- Although the study's originality lies mainly in its epidemiological approach to an understudied population, we will reinforce scientific rigor by ensuring that the methodology is described with maximum precision, allowing for reproducibility.
- The formulation of results has been revised to clearly specify T. gondii seropositivity as the primary outcome in all analyses, eliminating any ambiguity in interpretation.
- The discussion section has been enhanced to more robustly contextualize our findings within existing literature, explicitly addressing study limitations and proposing directions for future research. Hypotheses regarding higher prevalence in children, the absence of ocular manifestations, and the importance of the "garden factor," for example, have been more thoroughly discussed and substantiated.
We are confident that these in-depth revisions will significantly elevate the clarity, structure, and scientific rigor of the manuscript, making it a stronger and more valuable contribution to the literature. We thank Reviewer 2 again for dedicating their time and expertise to help us improve our work.

Reviewer 3 Report
Comments and Suggestions for Authors
This manuscript aims to investigate the risk factors and ocular health related to toxoplasmosis in quilombolas in the eastern region of the Brazilian Legal Ama-zon state of Tocantins Brazil. Although the topic is interesting in its scientific field, there are some issues that require the authors’ attention to improve the quality of this particular manuscript before further consideration for publication in a high-quality journal “Microorganisms”.
Specific comments:
- In this study, only children aged 5–7 and adults aged ≥60 are included, completely omitting the middle age groups. Why the authors do not consider adolescents and adults for analysis?
- Why the “garden factor” is considered as high-risk? Please specify.
- How parental education level may affect a child’s infection rate? Please clarify.
- The study mentions that all children are positive for both IgG and IgM, but the results of PCR tests are negative. Please elucidate why all IgM-positive cases show negative PCR outcomes.
- This study only draws from 4 quilombo communities. How the authors can ensure that the results of this study are representative of other quilombo communities? Please justify.
Author Response
Comments 1:
[In this study, only children aged 5 to 7 years and adults aged 60 years or older were included, completely omitting intermediate age groups. Why did the authors not consider adolescents and adults in the analysis?]
Response 1:
The focus on children and the elderly follows WHO recommendations for priority groups. We focused on ages 5–7 years because this is the initial window of schooling, when visual demands increase, and teachers can observe signs of low visual acuity. At this age, there is a peak incidence of visual problems, and many cases are reversible if identified early (amblyopia/refractive errors) (www.emro.who.int).
The presence of macular or paramacular lesions can necessitate educational adaptations and recommendations for auxiliary teachers and low-vision resources to prevent learning impairment in these children. In Brazil, according to the Brazilian Council of Ophthalmology, approximately 20% of schoolchildren have some ophthalmological alteration; ~10% require correction, and ~5% may experience significant vision reduction. Since young children cannot express that they have visual difficulties, early screening guides timely treatment, improves prognosis, and prevents psychomotor, academic, and social impacts. The elderly population also receives priority in WHO and Brazilian Ministry of Health ocular health assessment programs. Verifying the presence of toxoplasmosis-related chorioretinal lesions in these patients also helps to guide the type of approach to be taken for utilizing residual vision and improving quality of life in all social, professional, and family aspects (eoftalmo.org.br).
Comments 2:
[Why is the "garden factor" considered high risk? Please specify.]
Response 2:
The “garden factor” is considered high risk for toxoplasmosis, as demonstrated in the study for children (OR: 3.59) and the elderly (OR: 4.73), due to its direct association with environmental contamination by Toxoplasma gondii. Cats, being the definitive hosts of the parasite, eliminate millions of oocysts in their feces into the soil, and these oocysts can remain viable for up to 18 months, especially in moist soils such as those found in gardens and home vegetable gardens. The presence of a garden, therefore, creates an environment conducive to infection, as it allows direct contact between humans (especially children who play on the ground and adults who cultivate the land) and contaminated soil, or indirect ingestion of oocysts through harvested vegetables and fruits that have not been properly sanitized, or even through contaminated irrigation water, constituting a primary route of transmission of the disease. This factor is justified in lines 456 to 457.
To address this concern, we added the following text: “Gardens pose a high risk because cats, definitive hosts of T. gondii, shed viable oocysts for long periods in their feces in moist soil.”
Comments 3:
[How can parents' education level affect a child's infection rate? Please clarify.]
Response 3:
The parents' level of education is a significant risk factor for toxoplasmosis infection, mainly due to a lack of awareness about the transmission routes of Toxoplasma gondii and preventive practices. Parents with lower educational levels may have less access to or understanding of information about food hygiene, safe soil handling, the importance of hand hygiene, proper cat waste management, and the need to cook food thoroughly. Consequently, the adoption of protective behaviors in the home environment and in childcare may be impaired. In addition, low educational attainment is often linked to socioeconomic vulnerability, which can lead to poor sanitation, limited access to treated water, and practices such as home gardening (also a risk factor), where contact with the parasite in the soil is more likely. Thus, parental education acts as a crucial mediator in the implementation of protective measures, and its absence can expose children more directly and for longer periods to environmental risk factors present in quilombola communities.
To address this concern, we added the following text: This factor is justified in lines 428 to 443. “Low parental education levels influence childhood infection by limiting awareness of T. gondii transmission routes and preventive practices. This can result in poorer adherence to hygiene habits and proper environmental management, increasing children's exposure to risk factors in the home environment.”
Comments 4:
[The study mentions that all children tested positive for IgG and IgM, but PCR tests were negative. Please explain why all IgM-positive cases yielded negative PCR test results.]
Response 4:
The observation that cases with positive IgM serology had negative PCR test results is an important and clinically understandable finding in the context of Toxoplasma gondii infection. To understand this apparent discrepancy, it is crucial to consider what each type of test detects:
The presence of IgM usually indicates a recent or acute infection. These antibodies are the first to be produced by the immune system in response to the parasite and can persist for weeks, months, and in some individuals, even for more than a year, even after the active phase of infection has ceased.
PCR detects the genetic material (DNA) of the T. gondii parasite. A positive PCR result in peripheral blood samples indicates the active and circulating presence of the parasite at the time of collection.
Therefore, when IgM is positive and PCR is negative, this suggests that individuals may have gone through the acute phase of infection some time ago, resulting in the production of IgM antibodies that are still detectable. However, the parasite may no longer be actively circulating at detectable levels in the blood (which would be captured by PCR), indicating that the infection is in the process of resolving, or has already progressed to a chronic/latent phase, where the parasite encysts in tissues (such as muscles, brain, or eyes) and is not present in the bloodstream.
In short, the absence of a positive PCR for all IgM-positive cases indicates that, although there was recent exposure or reactivation that triggered the IgM response, the parasite load in the peripheral blood at the time of collection was no longer sufficient to be detected, or the infection had already transitioned to a non-active phase in the bloodstream. This is common in toxoplasmosis, which often becomes a latent infection after the acute phase.
Comments 5:
[This study is based on only 4 Quilombo communities. How can the authors ensure that the results of this study are representative of other Quilombo communities? Justify your answer.]
Response 5:
The authors of the study cannot “guarantee” that the results are representative of all quilombola communities in Brazil due to the great diversity of these communities in geographical, cultural, socioeconomic, and environmental terms. As the manuscript itself points out in the Discussion section, quilombolas have a history of high socioeconomic vulnerability, low family income, low educational levels, and limited access to basic sanitation and health services, characteristics that can vary significantly between different regions of the country. However, the study is of great value in focusing on a specific group in the eastern region of the Brazilian Amazon, in Tocantins, and in identifying risk factors (such as cat ownership, contact with soil, and parental education level) that are widely recognized as relevant to the transmission of toxoplasmosis in vulnerable populations. Thus, although unrestricted generalization is not possible, the results are highly relevant and applicable to quilombola communities that share similar socioeconomic and environmental characteristics to those studied.
The manuscript also highlights that “This was the first study we are aware of that evaluated exposure to T. gondii among quilombolas,” which positions it as an important pioneering study, providing a solid basis for future investigations in other communities and for the development of targeted public health strategies. The absence of a guarantee of universal representativeness is an inherent limitation of studies with localized samples, but it does not diminish the importance and applicability of the findings within similar contexts.

Round 2
Reviewer 2 Report
Comments and Suggestions for Authors
1.The title “Risk factors and ocular health related to toxoplasmosis in Quilombolas in the eastern region of the Brazilian Legal Amazon, state of Tocantins, Brazil” should be revised to:
“Risk factors and ocular health related to toxoplasmosis in Quilombola communities”
2.After the first mention of “Quilombolas in the eastern region of the Brazilian Legal Amazon, state of Tocantins, Brazil,” please use “Quilombola communities” consistently throughout the text. Replace all instances of “quilombos” and “Quilombolas” with “Quilombola communities.”
3.In the abstract, please spell out the full terms when IgG, IgM, ELISA, PCR, OR, and CI are first mentioned, and use the abbreviations thereafter.
Reviewer 3 Report
Comments and Suggestions for Authors
No further comment.
Author Response
We appreciate your time and effort spent reviewing and correcting our manuscript